# Conceptualizing the Role of Social Entrepreneurial Orientation in the Triple Bottom Line in the Social Enterprise Context: Developing Country Perspective

Madhuwanthi Premadasa * , Janaka Siyambalapitiya, Kumudu Jayawardhana and Imali Fernando

Faculty of Management, Uva Wellassa University of Sri Lanka, Badulla 90000, Sri Lanka; janaka@uwu.ac.lk (J.S.); kumudu@uwu.ac.lk (K.J.); imalif@uwu.ac.lk (I.F.)
* Correspondence: rafom@uwu.ac.lk

**Abstract:** Social entrepreneurship is becoming widely recognized as essential to developing economies and societies. However, we find that the lack of a clear and cohesive conceptualization for understanding the distinctive context and reliable role of social entrepreneurship is a challenging aspect. Furthermore, the research is lacking in developing country perspectives. Accordingly, this study argues that a social entrepreneurship conceptual model needs to be developed from a developing country perspective to advance the literature on the social entrepreneurship notion. Specifically, this study followed a qualitative research approach and conducted in-person semi-structured interviews with 24 Sri Lankan social enterprises by adopting the maximum variation sampling technique. The data were analyzed using thematic analysis. Consequently, our model explicates how social entrepreneurial orientation leads to the triple bottom line through dynamic capability and social innovation. The proposed model anticipates that social entrepreneurial orientation alone cannot achieve the triple bottom line in social entrepreneurship. Thus, based on existing research evidence, we believe that the following constructs—social entrepreneurial orientation, dynamic capabilities, social innovations, and the triple bottom line—can be integrated to provide a solid conceptual model for social entrepreneurial ventures in developing countries.

**Keywords:** social entrepreneurship; social entrepreneurial orientation; social innovation; dynamic capabilities; triple bottom line; developing country perspective



## 1. Introduction

Social entrepreneurship (SE) has recently gained prominence as a key segment of entrepreneurship. The unique mission and the market expectations are the key differences between social and commercial entrepreneurship [1]. The concept of entrepreneurial orientation (EO) has grown increasingly important in the entrepreneurship context [2]. The extant literature agrees that entrepreneurial orientation is a term established in the strategy-making process literature that represents firm-level entrepreneurship [3]. The most common dimensions to define EO in the extant literature are proactiveness, innovativeness, and risk taking [4]. The EO concept has been expanded into a variety of areas [5] and is a prominent and commonly utilized theoretical framework in management [6].

Social entrepreneurial orientation (SEO) originated as a challenge to the conventions of traditional business activities. Scholars have revealed that SEO has the overarching goal of achieving social impact, with SEO behaviors aimed at providing solutions to societal issues [7,8]. Here, social enterprises require creative and innovative ways to provide solutions to demanding social issues [9]. Further, scholars have asserted that exploring and exploiting an accumulated base of resources and competencies and developing new things are the two key challenges faced by any social enterprise. Accordingly, dynamic capabilities have been defined as a transformation of resources and competencies [10]. The dynamic capability approach illuminates innovation through the lens of the resources

and competencies of an organization [11]. Meanwhile, stakeholders have the ability to shape the development of an innovation because of their importance and the resources, access, and/or legitimacy that their support would bring [12]. Hence, the development of innovations and solutions that advance the public good is the main emphasis of social innovation. In fact, the key purpose of social innovation is to build a better future [13].

However, the extant literature offers fertile ground for further inquiry since four (04) subsequent important knowledge gaps can be observed. First, despite the growing recognition of this triple bottom line in achieving performance, this aspect has not received enough academic scrutiny in the social enterprise research context [14]. As discussed, traditionally, it is believed that it exists for dual value creation—social and financial—and there is a timely need to expand this traditional view of value creation with a triple bottom line paradigm of value creation. Second, current studies lack an illustration of how social entrepreneurship provides viable ground to promote social enterprises through achieving the triple bottom line: people-focused outcomes, financial-focused outcomes, and social-mission-based outcomes [14]. Most of the research has been limited to defining the concept and addressing conceptual ambiguities, differentiating social entrepreneurship from the mainstream of commercial entrepreneurship [15,16], and examining the value creation process. While there are several studies on the factors driving SE to accomplish its goals from varying perspectives, the knowledge is still generally dispersed and fragmented [17]. Third, there is a knowledge gap in understanding how social entrepreneurship drives the process of achieving these triple bottom line goals and in identifying what the antecedents and drivers of the process are [18]. Meanwhile, the definition of social entrepreneurship varies within the context of different geographical regions. Leaving definitional disagreements aside, a recurring theme in the literature on social entrepreneurship is that it is quite similar to how standard concepts of entrepreneurship are thought of [9,19]. Therefore, re-defining the conceptual boundaries of the respective constructs while identifying key drivers and antecedents of the process is time-valued. Fourth, the extant body of knowledge does not offer a comprehensive approach or model for top-level managers to understand what kind of effect their social entrepreneurial posture can have on achieving multiple dimensions of performance. Interestingly, this is still a blurred area in the extant body of knowledge that requires the immediate attention of current business researchers [20–22].

In light of the variety of historical and contextual elements, the SE phenomenon is viewed differently among nations and regions [23]. As a result, the concept of social entrepreneurship has been localized to reflect local practices. Further, most of the theories and definitions of social entrepreneurship have been formed based on the developed country perspective, which makes it difficult for developing country viewpoints to comprehend SE [24]. Economic, political, and social issues are particularly severe in developing nations. SE appears to be a successful strategy for finding innovative ways to address these economic and social problems. Thus, the notion of social entrepreneurship is evolving quickly and gaining yet more attention from policymakers and entrepreneurs in developing nations [14].

As a developing country, Sri Lanka is an island with high biodiversity. Roughly 200 years ago, 90% of the land was covered in forests. However, the rate of deforestation sharply increased with rising industrialization [25]. In the 21st century, the remaining natural forests are at risk, and national forest land continues to be eroded [25]. Consequently, the "Earth Restoration" in Belipola is the world's first analog forest that exhibits forest structures and functions that are comparable to those of a local natural forest catered to decrease deforestation in Sri Lanka. On the other hand, services for fundamental necessities such as health and education are limited, ineffective, or of poor quality in developing nations [14]. In Sri Lanka, non-communicable diseases (NCDs) such as heart disease, stroke, cancer, diabetes, and chronic lung disease account for more than 80% of deaths [26]. This serious issue was recognized by "Sehani Deshiya Oushada (Pvt) Ltd. (Buttala, Sri Lanka)", which produces valuable organic herbal crops and their value-added products for the Sri Lankan market, with the purpose of reducing the number of patients with

non-communicable diseases. Poverty reduction is a challenging and complex problem in many developing countries such as Sri Lanka and in the South Asian context as well. By giving them opportunities to generate revenue, the SE concept directly empowers those who are impacted by poverty. Thus, social enterprises can be used to provide sustainable solutions to social issues since they have the potential to generate both social and financial returns, which are more relevant to developing countries [27]. Thus, it is proved that many social enterprises simply strive to do their part in creating a better world, while some social enterprises' entire aim is centered around protecting the environment in the developing country context. However, social enterprises aim to generate a profit and expand their business in the same way that for-profit businesses do. Profit, though, helps them to achieve their ultimate goal of expanding their social purpose. Accordingly, it is highlighted that social enterprises and sustainable businesses can play a major role in achieving the triple bottom line [28,29].

Thus, SE must be conceptualized in the context of the larger and ever-changing competitive marketplace in which it works [30]. Therefore, SE should be conceptualized consistently to aid methodological advancement and to advance the realistic measurement of the notion. Thus, this study capitalizes on the four aforementioned important knowledge gaps, and the overall research objective of this study is to propose a conceptual model to theorize the role of social entrepreneurial orientation in the triple bottom line from the developing country perspective.

## 2. Literature Review

### 2.1. Entrepreneurial Orientation Literature in the Social Enterprise Context

As one of the emerging fields of study, entrepreneurial orientation (EO) has received significant attention [31] and has become a central concept in the entrepreneurship domain [1]. Thus, entrepreneurial firms have highlighted EO as a critical competency since it is seen as a requirement of their ability to identify and exploit opportunities in order to create value [32]. The author of [33] is one of the pioneers in developing the EO construct, consisting of three dimensions, namely (a) innovativeness, (b) proactiveness, and (c) risk taking, which have fostered multiple measurement scales and jointly define the process of entrepreneurship [3,33]. Entrepreneurial action heavily relies on innovativeness since it serves as the foundation for creating innovative business ideas [34]. First, ref. [33] conceived the idea that innovativeness is the willingness to do something new and creative through experimentation, resulting in unique or enhanced products, services, or processes. Thus, the ability to recognize and respond to possibilities and solutions is a key characteristic of innovativeness. Second, proactiveness has been defined as the capability of an enterprise to seek opportunities and practice forward-looking behavior in order to exploit market opportunities in a conscious effort to sustain itself in the market while competing with other businesses [33]. Thus, proactiveness relates to how organizations pursue commercial opportunities. Third, risk taking is the degree of willingness of managers to make significant and uncertain resource commitments. It includes the propensity to take bold actions such as moving into new, unknown markets, investing a large number of one's resources in questionable endeavors, and/or borrowing heavily [35]. EO has established itself as a distinct field within the entrepreneurship discipline, and academic studies in this area are extensive [36,37]. Despite the fact that social entrepreneurship is garnering significant attention in practice and research, academic studies on SEO remain relatively scarce [38,39].

Hence, SEO has emerged with the underlying motive of providing innovative solutions as it seeks to face societal challenges [7,8,40]. The combination of social and economic performances distinguishes SEO from activities driven by purely economic or social goals. As traced in the literature, SEO is a dynamic construct that is in its nascent stage [41], and simply, it is the effort of a social enterprise to build entrepreneurial orientation [42]. Based on the study conducted by [43], social innovativeness, social risk taking, social proactiveness, and socialness have been identified as the defining dimensions. This is based on the entrepreneurial orientation dimensions, namely innovativeness, risk taking,

and proactiveness, as identified by [33], who have pioneered the development of the entrepreneurial orientation dimensions. SEO is a combination of entrepreneurial behavioral dimensions, namely innovativeness, proactiveness, and risk taking [30]. Here, risk-taking behavior is replaced by the construct of risk management, as social entrepreneurship focuses more on the opportunity for social impact regardless of financial viability. Moreover, innovativeness, proactiveness, risk management, and social mission orientation have also been identified as dimensions of SEO.

### 2.2. Dynamic Capabilities in Social Enterprise Context

Most of the definitions of dynamic capabilities indicate that they are valuable [44]. A few other scholars have specified that dynamic capabilities create value indirectly [10]. However, if dynamic capabilities rely on social entrepreneurship, which earns profits, without which it would be linked with a strategic issue without considering the cost accounting approach, the strategic process may lead to high consideration of intangible assets [45]. Dynamic capabilities are defined as a firm's ability to produce, grow, and change its foundation of tangible or intangible resources and competencies [46]. Further, they have been defined as the capabilities of an organization to develop, modify, and improve its resources and competencies in the forms of tangible and intangible [46]. Thus, the heart of dynamic capabilities is the evolution of resources and competencies.

A prior study revealed that social entrepreneurship plays a crucial role in renewing competencies through dynamic capabilities [47]. Thus, the study concluded that there is a clear correlation between social entrepreneurship and dynamic capabilities. Further, it was revealed that social enterprises possess dynamic capabilities to create value for the community or ecosystems while also delivering long-term solutions to long-standing societal challenges [47]. Further, scholars have asserted that exploring and exploiting an accumulated base of resources and competencies and developing new things are the two key challenges faced by any social enterprise. One scholar has defined dynamic capabilities as a transformation of resources and competencies [10]. Thus, the dynamic capability approach illuminates innovation through the lens of the resources and competencies of an organization [11]. Study findings have revealed that Teece's dynamic capacities are, in reality, steps in the process of organizing social innovation [48]. According to Teece, developing a dynamic capability requires the ability to sense, seize, and transform [45]. These stages can be classified based on the nature of dynamic capability, displays of social innovation, and dominating the organizational process. Thus, dynamic capabilities could drive social innovation.

### 2.3. Social Innovation

In the 21st century, many disciplines have been using the "social innovation" concept, and it simply refers to the configuration of social practices that respond to systemic problems that aim to improve social well-being outcomes, including the participation of actors in civil society [49]. Accordingly, innovations can be recognized as a vital component of social entrepreneurship, as entrepreneurs must participate in innovation to provide solutions for the contemporary social issues that they want to tackle [50]. Recently, social innovation has gained significant attention from both scholars and policymakers as a valuable tool that can be used to help the community tackle complicated and complex social issues [51]. It is evident that social innovations efficiently generate employment opportunities, community services, and shared resources and empower the community. Thus, they will increase the capability of the people to face challenges in a successful manner.

Recently, academics have proposed different social innovation typologies [52,53]. The typology presented by [52] is more applicable to social enterprises as it explains the organizational perspective of social innovation and presents three levels of social innovation, namely incremental innovation, institutional innovation, and disruptive innovation. Incremental innovations relating to products and services are aimed at meeting social needs, and these innovations appear during the starting phase of a social enterprise [48]. Institutional

innovation denotes institutional change, and its purpose is to rethink social and economic structures to create social values [4]. Disruptive innovation is aimed at changing the whole system and envisions a shift in power dynamics and a reshaping of social hierarchies to the benefit of marginalized communities [52].

Social innovation is merely linked with social entrepreneurship and intrapreneurship [54], while some other scholars believe that social innovation can be frequently used interchangeably with social enterprises and social entrepreneurship [55,56]. Several authors have tackled concerns of social transformation using concepts of social innovation and social entrepreneurship, despite the academic heritages of their underpinning fields, innovation, and entrepreneurship [57,58]. As change agents, social entrepreneurs use systemic innovation to bring about a shift in the societal equilibrium [59]. In [60], social entrepreneurship was identified as a set of innovative actions and procedures that are used to identify, define, and exploit possibilities to increase social wealth by starting new businesses or reorganizing current ones. Thus, social entrepreneurship exists within a social innovation system that supports addressing social issues while shaping society.

*2.4. Social Entrepreneurship and Triple Bottom Line*

The triple bottom line is a notion that is strongly tied to the concept of sustainability. Sustainable development is defined as development that meets current demands while not endangering the ability of future generations to meet their own needs [61]. Many scholars have thus far attempted to include social enterprises in the structure of "hybrid organizations", which indicates that they create dual value: social value for their target groups and financial value to stay long-term financially sustainable [62–67]. However, the evolving for-profit literature suggests that organizations should engage in securing the triple bottom line, which states that organizational achievements could be three-fold: financial, social, and environmental.

First, the financial aspect of the triple bottom line is a variable that moves with the flow of the economic factors of any enterprise. Profitability, or personal financial gain, is the fundamental motivation for business owners [68]. In other ways, it refers to the capacity of the economy to withstand challenges and evolve in the future to provide for future generations as one of the subsystems of sustainability [69]. Thus, it focuses on the economic value created by an organization for its surroundings to prosper and enhances the potential of the organization to support future generations. Second, the social aspect of the triple bottom line relates to conducting business in a way that is fair to labor, human resources, and society. Simply, it is thought to add value to society. Thus, social performance focuses on an organization's engagement with the community to address problems related to community involvement, employee relations, and fair compensation [70]. Third, the environmental aspect of the triple bottom line is concerned with the effective utilization of environmental resources that does not compromise those resources for future generations. It entails the efficient utilization of energy resources, minimizing greenhouse gas emissions, and lowering the environmental impact [70]. Thus, the triple bottom line provides a framework to measure the performance and success of an organization in terms of financial, social, and environmental aspects.

The link between social entrepreneurship and achieving the triple bottom line is particularly important for any country as it assures the development of the economy [71], and social entrepreneurship has become one of the social, economic, and cultural global phenomena along with its rapid growth. Thus, social entrepreneurship could be able to balance its mission to create value for society while achieving financial sustainability [72]. The key aims of social entrepreneurship are to address the needs of the community and to serve disadvantaged communities while creating job opportunities. However, past studies have also revealed that it is difficult to balance these two competing objectives, namely social objectives and economic objectives. This task bias has arisen due to the excessive attention directed toward economic goals, which has led to social goals being ignored [73].

Most social entrepreneurship research studies examine the relationship between social entrepreneurship and sustainable development [74,75]. Meanwhile, the idea and implementation of the triple bottom line concept have also been discussed by previous scholars [76–78]. Further, studies have concluded that there is a requirement for a strong sustainable tool, which can be met by integrating the triple bottom line concept [79]. Thus, it is highlighted that social enterprises and sustainable businesses can play a major role in achieving the triple bottom line [28,29]. Accordingly, several studies have conceptually examined the relationship between entrepreneurship and the triple bottom line, and the majority of them have revealed a substantial positive link between the two factors. Thus, SE is defined by a lack of theoretical constraints, contradictory definitions and conceptualizations, gaps in the literature, and a scarcity of empirical evidence.

## 3. Materials and Methods

Considering the nature of the research phenomenon, this study used a qualitative research paradigm [80]. The purpose of qualitative research is to gain a thorough insight into a particular organization instead of providing a cursory picture of a wide sample of a population [81]. Initially, the researchers conducted semi-structured in-depth interviews with owner-managers or senior managers of 24 Sri Lankan social enterprises. In-depth interviews give researchers a lot more leeway in gathering data to support the intended research phenomenon [82]. Moreover, data were gathered from published research articles, annual reports, and website information. Further, the sample was obtained by using the maximum variation sampling technique since this allows for determining the maximum variation in social enterprises while recognizing their similarities, dissimilarities, and trends [83]. The rationale for using this sampling technique is supported since the scope of operations, size, target markets, products or services, and annual income of social businesses operating in Sri Lanka vary greatly.

Thematic analysis was used to analyze the gathered data in order to produce accurate and insightful findings. Generally, researchers must first become familiarized with the depth and breadth of the transcriptions, and thematic patterns must be developed through the identification of different codes. This study combined pure inductive and deductive thematic analyses to get maximum use out of the existing literature while establishing new themes based on the collected data.

## 4. Results

### 4.1. Propositions for Advancing Social Entrepreneurial Orientation in Achieving the Triple Bottom Line

The proposed framework explicates how SEO leads to the triple bottom line through dynamic capability and social innovation. The dynamic capability approach illuminates innovation through the lens of the resources and competencies of an organization [11]. Meanwhile, social enterprises need to come up with new and inventive approaches to addressing pressing social concerns [9]. Accordingly, the following conceptualization anticipates that SEO alone cannot achieve the triple bottom line in social entrepreneurship. Thus, based on the existing research evidence, we believe that the following constructs, social entrepreneurial orientation, dynamic capabilities, social innovations, and the triple bottom line, can be integrated to provide a solid conceptual framework for studying the triple bottom line of social entrepreneurial ventures (Figure A1).

### 4.2. Social Entrepreneurial Orientation

In [84], it was opined that SEO is a multidimensional phenomenon that involves the expression of entrepreneurially virtuous behavior to achieve a social purpose. The popular dimensions that the extant literature suggests define SE are proactiveness [85], innovativeness [86], risk management [87], socialness [24], and effectual orientation [87]. When it comes to tackling societal difficulties, SEO has emerged with the goal of offering creative solutions [7,8]. Social entrepreneurship is a set of innovative actions and procedures

that are used to identify, define, and exploit possibilities to increase social wealth by starting new businesses or reorganizing current ones. Accordingly, social entrepreneurship exists within a social innovation system that supports addressing social issues while shaping society. Social entrepreneurship plays a key role in modernizing competencies through dynamic capabilities [47]. Additionally, it was shown that social enterprises have the dynamic capability to benefit communities and ecosystems while also providing long-term solutions to pressing societal problems. Thus, SEO could lead to dynamic capabilities. A study defined dynamic capabilities as a transformation of resources and competencies [10]. Thus, the dynamic capability approach illuminates innovation through the lens of the resources and competencies of an organization [11]. Hence, dynamic capabilities could drive social innovation.

We propose the argument that SEO drives social innovation to achieve the triple bottom line and dynamic capabilities to develop social innovations. Thus, we propose the following:

**Proposition 1.** *SEO positively impacts social innovation.*

**Proposition 2.** *SEO positively impacts dynamic capabilities.*

**Proposition 3.** *Dynamic capabilities impact social innovations.*

Accordingly, this study proposes seven dimensions of SEO in the social enterprise research context from a developing country perspective, as follows.

<u>Innovativeness</u> refers to generating innovative ideas through the development of a new product or service and establishing technological leadership through the research and development of new procedures [88]. It also suggested that innovativeness plays a critical role in entrepreneurship, generating values that businesses desire while bringing about fresh modifications or combinations through innovative thoughts [89]. Due to the multidimensional origins of social problems, social entrepreneurs have various potential ways to exercise the tools or strategies of innovation to achieve their social mission. Hence, social enterprises have an opportunity to take a more proactive part in developing novel solutions to some of the most challenging issues facing the world today. One informant stated the following: " . . . *If they have capacity and innovative mind set, they will able to come up with innovative and novel solutions. This is the actual situation of the entrepreneurs in our country. Projects needed to prepare based on the innovative ideas of entrepreneurs and not based on pre-decided budgets . . .* ". Thus, social entrepreneurs must adopt a new way of thinking in order to bring about social change for the betterment of society [90]. Another social entrepreneur stated the following: "*we should provide our own product; not what market is demanding. Actually, people don't want this cup to drink tea. They can use even a coconut shell for that purpose. But people think to buy our product. Accordingly, I don't produce the product which is wanted by the society. I just produce what I want. However, the society will be moved towards me at a point. So, that is my talent and my skill. We always give a new product to the customer. If we are affluent in terms of knowledge and attitudes, we may be able to supply the products before the community demand*". Here, it is proved that innovativeness in social enterprises is different from that in traditional business enterprises. Commercial and mission-driven innovations, as well as creative approaches to combining the two viewpoints, are all included in innovativeness in social enterprises. Thus, we defined innovativeness as the willingness and imagination to advance new solutions for challenging social and environmental issues while sustaining commercially viable enterprises.

One of the key dimensions of SEO is <u>proactiveness</u>, which reflects the attitude to constantly pursue new opportunities [91]. Concerning social entrepreneurship, being proactive involves actively finding, analyzing, and taking advantage of social entrepreneurship opportunities. This is identified as a crucial factor of social entrepreneurship [30]. One respondent stated the following: "*majority of them cultivated peanut and they sold them to the nearest shop in their village. Those village sellers also act same and they tend to exploit those*

*female farmers as well as they did not receive fair profit for their production due to the higher level of involvement of the intermediaries. Based on those issues we formulated a new project in 2016. Then we requested from the Japanese organization to build a marketing center for us, request funds from them, clear mechanism to educate farmers to sell their harvest without the involvement of the intermediaries, machineries, data base to maintain information etc.*". Accordingly, this enterprise identified the probable challenges that would arise due to this issue in advance and acted accordingly. Another social entrepreneur highlighted the following: "*at the initial stage I decided to start this business after my graduation, but I thought, if I have sufficient time and technology, I can start this, 2 years prior to the expected time especially due to COVID-19 pandemic. If I start this business 2 years . . . prior to the expected time, then I can save my time and also, I can have good customer base when I will be completed by degree. Then I just have to focus only about the development side of my business*". Organizations in a dynamic environment benefit from proactive behavior because it helps them to obtain a first-mover advantage by responding to environmental changes. We found that this proactiveness illustrates two behavioral aspects of social entrepreneurs: how competent they are (a) to grasp the potential pains of social issues in advance and act accordingly and (b) to apprehend the potential commercial viability of the opportunity they are going to work on. Thus, proactiveness signifies the capability to recognize and respond to potential issues and difficulties.

*Risk management* is another key dimension of SEO, and many believe that being entrepreneurial involves exploiting while taking risks. Perhaps, there may be a difference in the level of risk taking between social and commercial entrepreneurs. Commercial entrepreneurs are eager to take on high-risk undertakings with lucrative rewards [92]. However, social entrepreneurs are cautious when committing resources to high-risk projects as they demonstrate risk management skills. One social entrepreneur explained the following: "*When Sarvodaya organization also asked about my opinions and that time I agreed to accept that offer. As a result of that contact, I was not permitted to continue my job. Therefore, I resigned from my previous job. I took the risk and then I managed everything in this field starting from that day. Now my products are available in super markets and currently, all supermarkets have this product.*" Social entrepreneurs exhibit risk-management skills rather than risk-taking ones. "*I am not making what people want, but the thing what I do is, I make what I do. It is a big risk. The benefit of that is, it leads to arise the hidden aesthetic person inside you. Even though you can't afford it, you just feel that product and try to buy it later. So, I target such points. Truly, I felt afraid to take the risk at the initial point. But I willingly manage that risk with the time. However, that fear lead me towards this.*" Social enterprises manage risks by recognizing them, mainly by scanning the environment, evaluating risk impacts, prioritizing risks, and monitoring, tracking, and controlling risks at an acceptable level. Accordingly, risk management reflects the tendency to identify risks, take controllable risks, allocate resources wisely, and plan prudently before committing capital to projects.

*Localness* is another key dimension of SEO, which fosters relationships with the community and culture, in turn helping to raise the value of local resources. Here, social entrepreneurs are more focused on the local social issues around them and are always eager to provide solutions. One respondent highlighted the following: "*there are plenty amount of Ginger and Lemmon grass plants in Kegalle area. Sometimes these plants are located at everywhere in-home gardens. But as we know there is no affordable market for these local things. There market price always up and down. No one can predict fixed income or revenue. Lots of farmers are not happy with this thing. They are searching for some cash earning way for better living. So, now I buy raw materials, I mean Ginger and Lemon Grass from local farmers, especially from Kegalle area. So, I can give a small help for them and able to take the use of our local resources*". Eventually, a social enterprise may attempt to take advantage of a natural resource opportunity to reach specific regions of resources and engage with the local community. Further, one social entrepreneur revealed the following: "*Another important one is we give a market value for banana tree. If people can make some money by using trunk of the tree, just imagine people will definitely move with Sri Lankan cultivations instead of foreign fruits. We have to focus on local things than foreign things. Corona situation teach us that. Do something creative like this one.*

*Actually, I was thinking about products which made by using local natural things. Mainly I need to do something by using local materials by adding value to them*". Sri Lankan social entrepreneurs utilize advantages taken from local resources to solve local issues in an innovative manner. Thus, localness reflects how sensitive social entrepreneurs are about the local social issues and the communities around them.

<u>*Social impact orientation*</u> is another dimension of SEO revealed in this study. Social impact orientation refers to an initiative that is designed and carried out with the goal of obtaining certain social results and impact. Desired improvements are expressed as specific social goals that orient and direct the overall work. One social entrepreneur stated the following: "*actually, we are not against men (laughing) . . . but, the majority is women . . . so it's a women-run . . . kind of thing. Since Women are the most affected by these differently-abled and blind-related issues. When it says . . . disable, it creates a more social stigma for women . . . blind women are affected more than men . . . it's normally happening in rural areas . . . So . . . it also affects . . . and, fifty-seven percent of those who are with disabilities are women. That's why we started this and how this evolved . . . now . . . *". Thus, a social enterprise may emerge with the intention of making a positive impact on the community. Further, some informants highlighted the following: "*we as a social enterprise emphasized on the positive aspects of providing equal rights for the females. We registered as a company in 2019 and currently work as a company with the aim of improving economic status of women's entrepreneurs and support them with social service activities. Moreover, we realized the importance of having adequate knowledge and economic empowerment to enhance the level of confidence of women. By considering those two aspects currently we continue to work with them. However, we are not going to stop our awareness programmes as well. So, our next step is to fight against obtaining the ownership of those lands where they conduct their farming from men. We are not going to talk about Masculinity, but we want to get same benefits for the women as well*". As a developing country, Sri Lanka is experiencing many social issues. Here, it is evident that organizations are initiated with the sole objective of providing a positive impact on women. Accordingly, we can define social impact orientation as the propensity of social entrepreneurs' behaviors to make an impact on an individual or a community as a result of their actions, activities, initiatives, or programs.

<u>*Value co-creation*</u> is another key dimension of SEO. The more diverse and connected the social network, the higher the probability that it will contain higher-value human capital qualities. This broadens the body of knowledge and creates space for creativity, improved problem solving, and practical planning to address societal problems [93]. The social entrepreneurs highlighted the following: "*I used my education knowledge and passion, and I started this ceramic organization. The main purpose of the organization is to improve the aesthetical feeling when they use these products. Because these products are unique. Another purpose is to give customers memorable products that can use long time. I create my own creations and I try to arise the hidden aesthetic person inside you. So, I do my passion to give the customer a memorable product*". Thus, it is a strategy for providing solutions and increasing value for all parties in which customers and enterprises make use of the interdependence of their respective resource bases. Additionally, the following was revealed: "*I had a passion for designing field our social enterprise produces furniture, storage containers, and housewares from natural, environmentally responsible materials and provides home-based work to women who cannot access other forms of employment. In addition to creating livelihood opportunities for low-income rural families, our social enterprise donates resources to support children with birth defects*". This denotes social entrepreneurs' behavioral orientation to consider the respective beneficiaries as an integral part of the solutions that they intend to design.

The final dimension identified in this study is *resource and capacity constellation*. In general, it may be stated that social enterprises have limited resources, including limited access to financial, technological, and qualitative and quantitative human resources [94]. Some people share the view that the primary problem of social enterprises is the lack of resources [95]. As a result, the Sri Lankan social entrepreneurs revealed the following: "*In addition, when we are arranging training programmes especially for women, we are joining at least one or two male participants for those groups as women cannot handle some tasks individually*

*during this process. At that time, they can get the help from those male participants in their groups*". Additionally, another stated the following: "*I already joined with Good Market. I am going every Saturday to Colombo for participate this event. From that I gathered lots of knowledge. New market trends, consumers need and wants, technology improvements and nature of the market. Not only that I am participating introductory exhibitions and fairs within Sri Lanka. I was able to gather lots of new things from those participations. I met new friends who doing social enterprises like me. With friendly discussions with them I got totally new ideas*". This is a reflection of how to allocate and rearrange internal and external resources and competencies to maintain the desired social objective. Accordingly, we also found that social enterprises, being highly constrained resource-wise in nature, show a trend of developing their unique approaches to gaining resources and capabilities that they cannot build in-house.

*4.3. Social Innovation*

We propose six dimensions of social innovation in the social enterprise research context from a developing country perspective.

*Market efficiency-based innovation* is one key dimension of social innovation, and here, social enterprises improve their economic and social performances by having the ability to comprehend the current market and continually take external market changes into consideration by undertaking radical product and service innovations. One of the social entrepreneurs highlighted the following: "*Then I proposed this, which is based on the water sports activities. It was able to do very good water sport activities, in Trincomalee district. With the google map, this is Trincomalee district. This is the JKAB beach resort, this one. This is JKAB park hotel. The other property is JKAB Lagoon in Nila Valley. My idea is wild life adventure. Here, is this beach, there is the boat service. Around 7 people can travel in this boat. There come from JKAB Beach to the pigeon island*". Through the introduction of new products and services or the satisfaction of unmet needs, social entrepreneurs become more efficient at engaging the market. Further, another informant stated the following: "*I use banana stems and pineapple bushes . . . . actually, after taking the harvests . . . to make them. I use fibers in them. We call that*" *Vegan Leather*". This provides a competitive advantage over one's rivals by being the first mover. Thus, market efficiency-based innovations focus on increasing the efficiency of the market engagement of social enterprises.

Another key dimension of social innovation is organizational efficiency-based innovation. It focuses on enhancing the internal processes of social enterprises to increase productivity and profitability. A respondent said the following: "*Therefore most of the time we use hand processing techniques. Other than that, we are using traditional equipment such as mortar to process our products. In addition, other than drier, we are giving less priority to the other machines in our production process. Drier is the most frequently using machine in this process. This is the production process and how the productions and tea processing are currently going on in my factory . . .* ". This is intended to increase the efficiency within the organization while reducing unnecessary financial and physical resources. Another informant highlighted the following: "*When they are come to weave for one or two months. When we are satisfied, we are asked to work from home. Because, while they look after children, family they can work. When we are getting satisfied, we are asked them to go home and do. We can't put all into one place. It will be very expensive. Thus, we will give them these weaving machines. We have one supervisor and a technician, they go home and looking them. If there is any mistake or anything happened, they call us. We will give all including yarns and designs*". Here, they really intend to reduce the wasteful human, financial, and physical resource spending of the enterprise to gain maximum use of them. Thus, organizational efficiency-based innovation focuses on enhancing inter-organizational efficiency.

Another important dimension of social innovation is mission efficiency-based innovation. The purpose of social enterprises is to create sustainable socio-economic structures, networks, institutions, organizations, and practices that create and maintain social benefits. One social entrepreneur revealed the following: "*we just offering a new product; a blend of 42 cereals instead of milk powder only within our outlet. This is very nutritious and it could be*

*able to obtain the quick results for diabetic. Meantime, this is good for everyone from the child aged 6 months to upward for all other diseases and it boosts the immunity as well*". Here, the main focus is on exploring the most efficient models of delivering the intended impact that can bring advantages for both social ventures and the target communities. According to another informant, "*usage of palm oil and imported unhealthy oil has created many issues. Due to unhealthy coconut oil and palm oil people living in our country have to face different type of diseases. Recently we can see the incident related to the imported coconut oil. As a solution for unhealthy coconut and palm oil we produce and distribute virgin coconut oil for reasonable prices to make people healthier*". By developing these innovations, social enterprises are primarily focused on carrying out organizational missions efficiently. Accordingly, based on this study, mission efficiency-based innovation focuses on efficiency when delivering social missions.

Based on this study, market effectiveness-based innovation is another significant dimension of social innovation. This entails experimenting with innovations to see how well they can interact with the target audience. Based on the responses, the following was highlighted: "*currently we mainly search internet for creating new knowledge and we try to identify the market opportunities. Through that, we have identified the current necessities of the market. Currently we are testing to innovate herbal sanitizers by using natural ingredients by using turmeric, kohomba, etc.*". This supports social enterprises in effectively reaching the target market with the required products. According to one informant, "*with that mind I found a package which made by using natural things. In Sinhala we called it as "Pankola Petti". I focus on innovative things and made a unique package for it*". It is easy to reach the expected market segment with a low cost. Hence, market effectiveness-based innovation focuses on incubating novel initiatives depending on how effectively they can actively engage with the target market.

*Organizational effectiveness-based innovation* is another dimension of social innovation. Organizational effectiveness depends on employee attitudes, operational effectiveness, and financial effectiveness. Some of the interviewees highlighted the following: "*Any one did not guide farmers to apply this strategy and they found that solution by own. We were aware those farmers regarding those moths and their life cycle, the periods they lay eggs and everything. So, employees found an innovative way to find the solution to this issue*". This leads to a reduction in organizational resource costs and supports the social enterprise in sustaining itself in the future. Further, another interviewee highlighted the following: "*I had a background that my brother is involved in a business related to machineries, I modified these machines with his support. We are developing new machineries and new techniques of designing we like to improve the quality of the product with new innovation and researches in a way that we can reach our market segment. That is, we modified the existing machine based on our* requirements". Social enterprises attempt to re-invent existing methods to increase organizational effectiveness. Thus, organizational effectiveness-based innovation focuses on raising the effectiveness of social ventures to sustain enterprises in the long run.

*Mission effectiveness-based innovation* is the final dimension of social innovation. In social enterprises, mission effectiveness is important, and it should concentrate on providing remedies intended to solve societal issues and give value to society. Here, an informant disclosed the following: "*I have made these pencils using old paper. We can use this as a normal pencil and even we can sharpen it as well. Mmmm . . . Then . . . we cut trees to produce both paper and pencils. So, to make a ton of paper we do a lot of environmental damage. I told you that I build concepts . . . Then if we make pencils using old newspapers we can protect the trees that are using to make pencils. That's the concept and the idea I'm trying to convey*". This highlights how social enterprises achieve effectiveness when delivering social missions. Another social entrepreneur highlighted the following: "*I am not depending on machines most of the time to produce those things . . . Therefore, most of the time we use hand processing techniques. Other than that, we are using traditional equipment such as mortar to process our products. It creates job opportunities and it is easy to achieve my social mission*". Social entrepreneurs aim to provide remedies intended to solve societal issues and give value to the community.

Therefore, this study highlights that mission effectiveness-based innovation focuses on raising effectiveness in delivering social missions.

*4.4. Dynamic Capabilities in Social Enterprises*

This study proposes three dimensions of dynamic capabilities in the social enterprise research context, as follows.

The capability to sense opportunities is the first dimension of dynamic capabilities, which has been highlighted as a critical action when dealing with dynamic capacities, requiring rigorous examination, formation, investigation, and explanation. The capacity of a business to recognize, understand, and exploit environmental opportunities [96] defines the capability to sense opportunities in the commercial enterprise sphere. The extant literature indicates that social entrepreneurs see opportunities and exploit them by being inspired by an ethical core [97]. Accordingly, the informants highlighted the following: "Good market participation, Exhibitions and career fairs, Ideas from consumers, Ideas from foreigners and Family advices and suggestions are helping to identify opportunities". Here, social entrepreneurs recognize and examine prospects through self-learning and acquire new knowledge in many ways. Another informant revealed the following: "before the COVID-19 pandemic, we only focus the souvenir and after the pandemic we identify an opportunity to housing items such as table clothes, bedsheets, wooden items, they always come up with the new ideas". During and after the crisis, Sri Lankan social enterprises discovered a plethora of new opportunities. It is shown that social enterprises regularly assess environmental changes to sense new opportunities. Thus, the capability to sense the opportunities in social enterprises is defined as the ability to scan and calibrate opportunities by interpreting the changes in the environment.

The capability to seize opportunities is the second dimension of dynamic capabilities. Enterprises seize what they have obtained through sensing actions to demonstrate their capacity to react to the environment [98]. SEs acquire and make use of resources to take advantage of opportunities to pursue their dual objectives [99]. An informant highlighted the following: "sometimes . . . You may have question about how I learnt those things . . . Actually . . . I went several factories . . . There is a factory called "Amba" in Bandarawela. I visited them and observed entire process. Then I thought about the things what I can add to this . . . I search them through internet . . . Actually, not the same thing what they produce . . . But . . . I wanted to do new thing by adding value. In that way I searched and learnt new things and then I trained my employees accordingly." Social companies frequently take advantage of networking opportunities and strengthen these networks to accomplish their social missions by building up the loyalty and commitment of their consumers and marketers [100]. External networks serve as a solid basis to help SEs understand and develop their decision-making process as they pursue social and financial objectives by grabbing opportunities. Another interviewee revealed the following: "Moreover, people who engage with Ayurvedic medicine practices are facing many difficulties to find bee honey with good quality. So, they use Wasp honey instead of Bee honey. Actually, they also can use quality bee honey if they focus on develop this and add value to it. However, people still follow those traditional methods. Hence, if we can change those traditional attitudes and believes of people, we have many opportunities and long way to move forward". Social entrepreneurs create structures, processes, and designs that will motivate social ventures to take advantage of opportunities, that is, to seize opportunities. Thus, the capability to seize opportunities in social enterprises corresponds to the ability to pursue possible opportunities and act quickly to take advantage of them for the benefit of society.

The third dimension of dynamic capabilities is the capability to manage threats through reconfiguring. Simply, this is the ability of an organization to combine and rearrange resources and organizational structures when the environment shifts [101]. Reconfiguration in social enterprises permits them to establish and maintain social missions through business model structuring, by combining resources to provide social value, and by creating new markets for new products [102]. Here, a respondent stated the following: "definitely.

The natural die, what I have previously told, we have doing many researches regarding this. We have to have a proper idea about the amount of die needs to add, while protecting the quality of the natural die prepared by using Veniwal Geta, turmeric, neem (kohomba), shoe flower (wada mal). Therefore, continuously it needs to undertake the sample checking to identify the suitability of those ingredient to make this fabric." Social enterprises manage threats by reconfiguring existing resources to satisfy changing customer needs discovered through research and development. A social entrepreneur disclosed the following: "we have to look at our internal things, how to improve? Bring more outputs, and the cost down likewise . . . . so we are doing those things internally to manage threats". This involves rearranging activities, resources, and enterprise skills internally while being adaptable and adjusting to new circumstances in order to handle threats. Thus, the capability to manage threats through reconfiguring in social enterprises refers to the ability to be flexible and adjust to changing factors, conditions, or environments and to manage threats while managing the tasks, resources, and competencies of the enterprise.

### 4.5. Social Innovation in Achieving the Triple Bottom Line

It is commonly understood that current Western levels of living cannot be sustained permanently and that a transition to sustainability necessitates dramatic alterations in our lifestyles. In [103], it is contended that social innovations are required to transition from the present unsustainable living models to new, more sustainable ones. Recently, social innovation and sustainability have become more prevalent as solutions to common problems. Social innovation allows for meeting unmet social needs while empowering the community and having them participate in sustainable growth. It has been identified as a key factor that leads to sustainable requirements, and a direct relationship between social entrepreneurship and sustainable development can be identified [104]. As a result, social innovation has gained attention among policymakers and academics in the last decade as a potential tool for assisting communities in addressing complex societal challenges and unmet social needs, driving them toward achieving the triple bottom line [51]. Thus, social innovation could drive enterprises towards achieving the triple bottom line. We propose the following:

**Proposition 4.** *Social innovation impacts the triple bottom line.*

Further, we propose three dimensions of the triple bottom line in the social enterprise research context.

The first dimension of the triple bottom line is the <u>*environmental*</u> dimension, corresponding to the efforts of an organization to limit its impact on the environment, as well as its usage of energy and waste production, in order to lessen its ecological footprint [105]. Environmental sustainability and more general objectives of social value creation could be examples of the extent to which social enterprises are socially integrated [106]. It evaluates the degree of environmental responsibility practiced by an enterprise. Here, one informant emphasized the following: "*my all products are made 100% with discarded papers, which can be washed and reused as well. My products are water resistance. They can clean by water when get some dirty look from them. Aims to promote recycling and reduce the use of plastic. We make crafts, greeting cards, and stationery from upcycled newspapers and other waste materials. There is no harm for environment as well. And I always concern on environment. With the university experiences, I thought about ecofriendly products which gives good impact for our environment*". Social enterprises are frequently required to choose between a less expensive choice and an environmentally friendly and sustainable alternative. Another respondent specified the following: "*the purpose of my social enterprise is promoting eco-friendly products in Sri Lanka. Not only that we are making polished coconut shell-related fancy items, pet collars, hand bags and traditional statues by using natural things. When talking about banana tree, after picking fruits there is no usage of it. People just cut down trees and them just sinking to the ground. But I think about some kind of a usage from that. After learning about extracting fiber from banana tree I try to*

*making products which can take good prices in the market. All of my works are based on save the environment concept and I try my best to make my all products without gaining any harm to the environment as well as society*". Enterprises try to protect the environment from which they obtain their raw materials. Thus, we define the environmental aspect of the triple bottom line as the protection of nature from which organizations obtain their raw materials and deliver their output.

The *social* dimension is the second dimension of the triple bottom line. The social dimension includes the effect of an enterprise on social welfare, considering both its employees and the community at large. Thus, it focuses on addressing topics such as educational help, social contact, charity causes, and fair pricing policies [105]. An interviewee highlighted the following: "*not only that by using this social enterprise, I buy raw materials I mean Ginger and Lemon Grass from local farmers. Especially from Kegalle area. So, my sole intention is to give a small help to them*". The social pillar of the triple bottom line highlights the most concerned, sophisticated part of any social enterprise. Another informant highlighted the following: "*the important thing is that my products are very popular among tourists. They really love it. I open some showcases in front of that hotel and they used to buy my banana fiber products. All revenues received are spent on the well-being of disabled people. I do not need profits from this work* Hence, social entrepreneurs are very concerned about the well-being of society rather than earning profits. Another interviewee emphasized the following: "*by starting this firm, we have generated many jobs opportunities, which makes women to stay at home with their family while increasing their living standard*". The social aspect addresses issues with community involvement, employee relations, and fair compensation [70]. Thus, this study defines the social pillar of the triple bottom line as being concerned about the issues in society and providing valuable solutions to them.

The *economic* dimension is the third aspect of the triple bottom line, which is common to all enterprises. Based on the extant literature, it corresponds to the ability of a business to make profits or reduce costs [107]. The economic aspect of the triple bottom line expresses the potential of the economy to survive and evolve in the future in order to support future generations. In the interviews, the following was highlighted: "*usually, we expecting profit like all other companies. So, we operate expecting profit as well as social welfare. Therefore, social welfare is also a key expectation and we focus on living standard of our employees specially women employees*". Social entrepreneurs reorient their attention from a narrow economic focus to broader social and environmental impacts. Further, an informant disclosed the following: "*Actually, the profit is not the main target or our hope. Women empowerment. In the future, another big factory will be opened. The main purpose of that is providing job opportunities to a huge community base. Because I don't think employees are not staying with us if we are only looking for profit*". Together, social enterprises try to achieve social and environmental objectives with financial gains through social entrepreneurship. However, it is evident that they earn profits to survive and that their ultimate goal is social welfare. Accordingly, we define the economic aspect of the triple bottom line as the internal financial stability and profitability that ensure the existence of a business.

*4.6. The Moderating Role of Stakeholder Pressure*

Meeting the requirements of a firm's stakeholders, whose support is essential in establishing legitimacy for the social entrepreneur's actions, is the greatest way to fulfil the dual purpose of social enterprise. The goal of stakeholder theory is to identify the pertinent stakeholders, as well as their interests, and to try to balance these interests that are frequently at odds when managers allocate and distribute resources [12]. Instead of being the sole product of social entrepreneurs' vision, social businesses are the result of the social, cultural, commercial, and political expectations of stakeholders concerning the spectrum of innovation [108]. Accordingly, it has been revealed that stakeholder pressure can impact the introduction of innovations due to their importance, as well as the resources, accessibility, and/or legitimacy that their support would provide. Accordingly, we propose the following:

**Proposition 5.** *Stakeholder pressure moderates the relationship between SEO and social innovation.*

Accordingly, this study proposes two dimensions of stakeholder pressure from a developing country perspective.

Internal stakeholder pressure is the pressure from internal stakeholders. The extant literature on internal stakeholder pressure has mainly focused on employees, managers, and shareholders [109]. A social entrepreneur disclosed the following: "one of the issues faced by this bee keeping is the death of moths. Those insects are attacking the outer layer of the bee hive and that layer becomes empty and it is like a net. I requested to suggest several solutions for this issue from the farmers. Actually, we had a farmer in the "Katharagama" area and that farmer was able to find a solution for this issue". Production-related choices are the main way that internal stakeholder influences organizational effectiveness. The management pressure its employees to suggest innovative ideas in order to achieve the predetermined social goals. Another informant said the following: "The hotel is operating with the solar power, the electricity. Sustainability is there. Another one is treatment plants. When you come my hotel, you can see a lot plant there. So, I started JKAB Landscape. During the period from March to April my all staff involved in nurseries making nursery plants and organic fertilizers. For an example, believe that around 2000 king coconut during that period little palms around 4000 numbers we prepared. Then I wrote a story to all employees and I made the arrangements the staff to involve this work". To maximize the effect of internal stakeholders, who are projected to have the largest social, economic, and environmental impact, social entrepreneurs should make every effort to organize their views and streamline their operational processes. Thus, this study defined internal stakeholder pressure as the pressure from a group of internal stakeholders to implement various social and environmental strategies and practices.

The second dimension of stakeholder pressure is external stakeholder pressure. Competitors, the media, regulators, the community, business partners, government entities, and suppliers are among the external stakeholders [109]. Primarily, pressure from external stakeholders is crucial for businesses as they have the ability to harm or help enterprises by promoting or imposing changes on current practices [109]. A respondent specified the following: "During three months March, April and May period, there was a huge new idea competition in Sri Lanka, South Asia and East Asia which is organized by a German NGO". The topic was "how you can generate income during COVID-19 period without terminating your staff. When you come to my hotel, you can see a lot plant are there. During that period from March to April my all staff were involved in nurseries making nursery plants and organic fertilizers. There I participated and then I wrote a good story. They have given me first place. (Happily) and received 1.5 million". There should be a good alignment between the expectations of external stakeholders and social concerns, which will determine the ability of a firm to trade its goods. The pressure from external stakeholders encourages social enterprises to adopt innovations. Another social entrepreneur highlighted the following: "moreover, NGOs helped us and they encourage us towards these social service activities. Further, they agreed to pay salary for about 3 or 4 members in our society as well. As a result, we formulated many women's societies and worked against social injustices and we registered our society in 2002 to legalize". External stakeholders make participation decisions that involve providing the organization with resources and contributing towards organizational effectiveness. Thus, external stakeholder pressure in social enterprises can be defined as the pressure from a range of external stakeholders to implement various social and environmental strategies and practices.

## 5. Conclusions, Implications of the Study, Limitations, and Future Research Directions

We argued that a social entrepreneurship conceptual model needs to be developed from a developing country perspective to advance the literature on the social entrepreneurship notion. Thus, this study proposed a conceptual model to theorize the role of social entrepreneurial orientation in the triple bottom line from a developing country perspective

to advance the literature on the social entrepreneurship notion by making a clear departure from the for-profit sector. It explicates how social entrepreneurial orientation leads to the triple bottom line through dynamic capability and social innovation. Further, it highlights the moderating role of stakeholder pressure in the relationship between SEO and social innovation. The dynamic capability approach illuminates innovation with the lens of the resources and competencies of the enterprise, while social enterprises need to come up with new and inventive approaches to address pressing social concerns. Accordingly, the proposed model anticipates that social entrepreneurial orientation alone cannot achieve the triple bottom line in social entrepreneurship. Thus, based on existing research evidence, we believe that the following constructs, social entrepreneurial orientation, dynamic capabilities, social innovations, and the triple bottom line, can be integrated to provide a solid conceptual model for social entrepreneurial ventures in developing countries. Accordingly, this study contributes to the ongoing discussion of developing context-specific theories in social entrepreneurship while providing significant implications for theory, practitioners, and policymakers.

Accordingly, we provide two contributions to the theory. First, this study contributes to the literature on SE by developing a conceptual model to theorize the role of social entrepreneurial orientation in the triple bottom line by rethinking the SE construct, which improves the understanding of the SE context. Second, we add to the discussion about developing context-specific SE theories that are currently taking place in the field. This study also has important practical implications for practitioners because it identifies the behaviors to foster in order to develop viable social enterprises. This study also has implications for policymakers by giving them valuable information about the factors they should consider while supporting and developing the SE sector in the Sri Lankan context.

There are very few constraints in this study, but they suggest important areas for future research. First, this study only reached out to a limited number of social enterprises. However, this study's sample plan was effective enough to capture a lot of variety within the industry. Future scholars can therefore confirm the findings with more empirical investigations. Second, future scholars might empirically test the model to strengthen the validity of the suggested conceptualization.

This study explicates how social entrepreneurial orientation leads to the triple bottom line through dynamic capability and social innovation. Notably, this research proposes ways for SE researchers to quantitatively validate the suggested SE conceptual model in the future. They should also consider how well the generated dimensions would fit into research on various social enterprises in various geographical areas across the world. Overall, by highlighting the significance of rethinking SE from the perspective of developing countries, this study contributes to the academic field of SE.

**Author Contributions:** Conceptualization, M.P., J.S., K.J. and I.F.; Methodology, M.P., J.S., K.J. and I.F.; Formal analysis, M.P., J.S., K.J. and I.F.; Data curation, M.P. and K.J.; Writing—original draft, M.P.; Writing—review & editing, J.S., K.J. and I.F.; Supervision, K.J. and I.F.; Project administration, and J.S. and M.P.; Funding acquisition, J.S. All authors have read and agreed to the published version of the manuscript.

**Funding:** This research was supported by the Accelerating Higher Education Expansion and Development (AHEAD) Operation (DOR-grant number 11) of the Ministry of Higher Education, funded by the World Bank.

**Institutional Review Board Statement:** Not applicable.

**Informed Consent Statement:** Not applicable.

**Data Availability Statement:** Not applicable.

**Conflicts of Interest:** The authors declare no conflict of interest.

## Appendix A

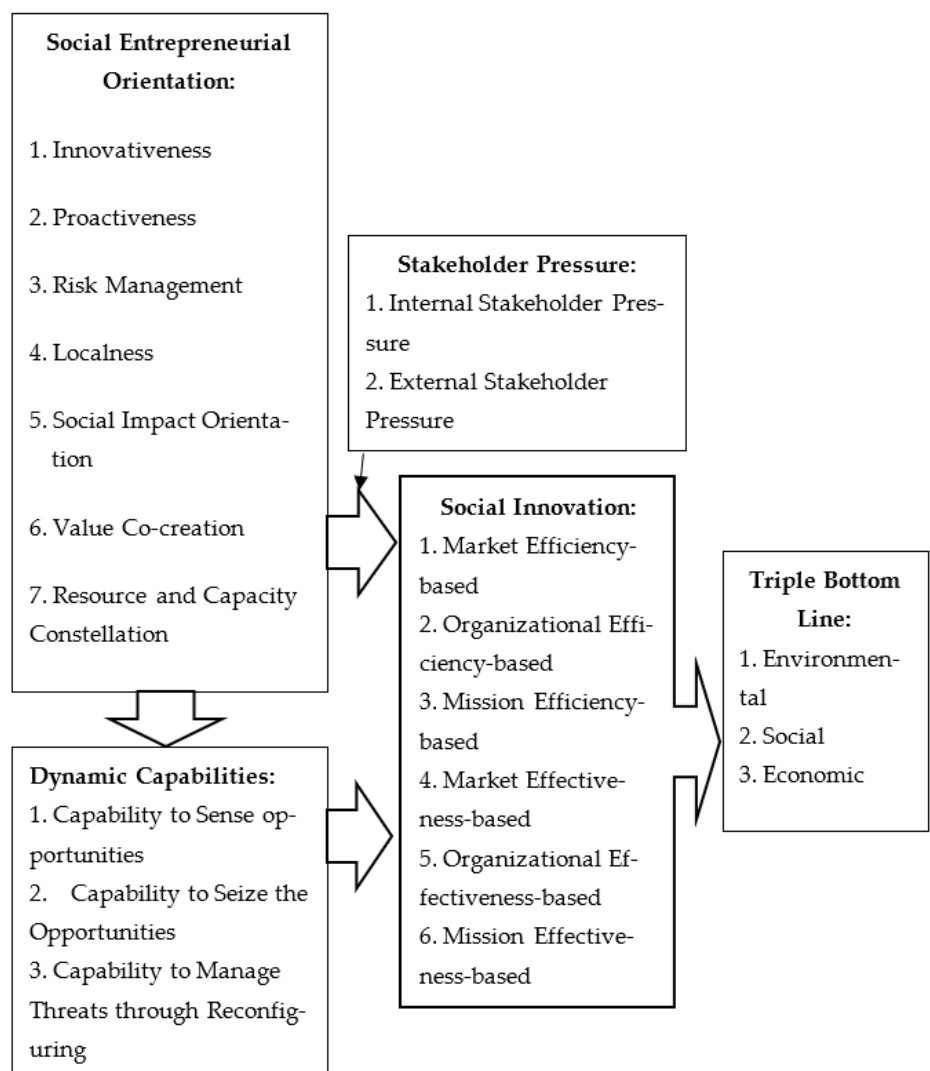

**Figure A1.** Conceptualization of Social Entrepreneurship in Developing Country Context.

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
