# Peer review of "Conceptualizing the Role of Social Entrepreneurial Orientation in the Triple Bottom Line in the Social Enterprise Context: Developing Country Perspective"

_sustainability, doi:10.3390/su15118759_

Round 1
Reviewer 1 Report
1. The abstract follows a logical structure and well written
2. The conceptual link between social entrepreneurship and the triple bottom line in the introduction section is weak and needs more explanation
3. I suggest adding contextual significance in the introduction section
4. There are vague statements by the author like line 32. The concept of EO. EO is a concept? or a strategic orientation of the firm?
5. Line 161, DC are valuable? where are the references who said and what were the contexts or framework or sectors they were mentioning? Authors need to explain the DC from Teece (1997) and then use more recent citations to elaborate on it
6. Line 172, prior literature… is there only 1 study claiming that?
7. In methodology, authors did not mention the methodological and analysis techniques and tools properly, how they were used, significance, and weaknesses
8. Arguments in thematic analysis need supporting or contradictory benchmark
9. The conclusion needs policy, managerial and practical implications and highlights the study's significance in paving way for future studies.
Author Response
Dear Sir/Madam,
Thank you for your valuable comments and those were supported to enhance the quality of our study. Herewith, we have provided the point-by-point responses to the reviewer's comments. Please find the responses.
1. The abstract follows a logical structure and well written.
A good comment.
2. The conceptual link between social entrepreneurship and the triple bottom line in the introduction section is weak and needs more explanation.
I explained the conceptual link between social entrepreneurship and the triple bottom line in the introduction section by using extant literature. The literature highlighted that social entrepreneurship could play a major role in achieving the triple bottom line.
3. I suggest adding contextual significance in the introduction section.
I added the contextual significance in the introduction section. Most of the theories and definitions of social entrepreneurship have been formed based on the developed country perspective, which makes it difficult for developing country viewpoints to comprehend SE. Economic, political, and social issues are particularly severe in developing nations. SE appears to be a successful strategy for finding innovative ways to address these economic and social problems. Thus, the notion of social entrepreneurship is important to be studied in the developing country context.
4. There are vague statements by the author like line 32. The concept of EO. EO is a concept? or a strategic orientation of the firm?
I corrected the statement in line 32.
5. Line 161, DC are valuable? where are the references who said and what were the contexts or framework or sectors they were mentioning? Authors need to explain the DC from Teece (1997) and then use more recent citations to elaborate on it.
I rewrite the line 161 and explained the DC by using the Teece model and other existing literature.
6. Line 172, prior literature… is there only 1 study claiming that?
I added literature here.
7. In methodology, authors did not mention the methodological and analysis techniques and tools properly, how they were used, significance, and weaknesses.
In methodology, it is mentioned that the study conducted semi-structured in-depth interviews with owner-managers or senior managers of 24 Sri Lankan social enterprises. Then thematic analysis was used to analyze the gathered data. First, the researchers familiarized with the depth and breadth of the transcriptions, and next, thematic patterns must be developed through the identification of different codes. Accordingly, this study combined pure inductive and deductive thematic analyses to get maximum use out of the existing literature while establishing new themes based on the collected data.
8. Arguments in thematic analysis need supporting or contradictory benchmark.
I supported the arguments in the thematic analysis with the previous literature. However, the newly discovered themes were supported mostly by the gathered data from the interviews.
9. The conclusion needs policy, managerial and practical implications and highlights the study's significance in paving way for future studies.
This study contributes to the ongoing discussion of developing context-specific theories in social entrepreneurship while providing significant implications for theory, practitioners, and policymakers. Finally, it provided the significance of the study while suggesting future research directions.
Reviewer 2 Report
Thank you for the opportunity to read the work. I work with an interesting subject but which I consider has a very important basic problem. The authors comment that a social entrepreneurship conceptual model needs to be developed from the developing country
perspective to advance the literature on the social entrepreneurship notion, but this has not been sufficiently argued or justified. Without doing an exhaustive search, several previous studies can be found in this regard:
- Salamzadeh, A., Azimi, M. A., & Kirby, D. A. (2013). Social entrepreneurship education in higher education: insights from a developing country. International Journal of Entrepreneurship and Small Business, 20(1), 17-34.
- Azmat, F. (2013). Sustainable development in developing countries: The role of social entrepreneurs. International journal of public administration, 36(5), 293-304.
- Nicholls, A. (2013). The social entrepreneurship–Social policy nexus in developing countries. In Social policy in a developing world (pp. 183-214). Edward Elgar Publishing.
- Del Giudice, M., Garcia-Perez, A., Scuotto, V., & Orlando, B. (2019). Are social enterprises technological innovative? A quantitative analysis on social entrepreneurs in emerging countries. Technological Forecasting and Social Change, 148, 119704.
- Haughton, A. (2013). Social entrepreneurship: Reducing crime and improving the perception of police performance within developing countries. International journal of Entrepreneurship, 17, 61.
- Satar, M. S. (2016). A policy framework for social entrepreneurship in India. IOSR Journal of Business and Management, 18(9), 30-43.
- Foryt, S., Lakshmi, U., & Kogut, B. (2002). Social entrepreneurship in developing nations. Research Paper, INSEAD, Fontainebleau.
- Littlewood, D., & Holt, D. (2018). Social entrepreneurship in South Africa: Exploring the influence of environment. Business & Society, 57(3), 525-561.
- Anh, D. B. H., Duc, L. D. M., Yen, N. T. H., Hung, N. T., & Tien, N. H. (2022). Sustainable development of social entrepreneurship: evidence from Vietnam. International Journal of Entrepreneurship and Small Business, 45(1), 62-76.
Author Response
Dear Sir/Madam,
Thank you for your valuable comments and those were supported to enhance the quality of our study. Kindly note that, we have responded to the reviewer's comments as follows. Please find the responses below.
01. The authors comment that a social entrepreneurship conceptual model needs to be developed from the developing country's perspective to advance the literature on the social entrepreneurship notion, but this has not been sufficiently argued or justified.
Now we have provided more justifications in the introduction section.
In light of the variety of historical and contextual elements, the SE phenomenon is viewed differently among nations and regions. As a result, the concept of social entrepreneurship has been localized to reflect local practices. Further, most of the theories and definitions of social entrepreneurship have been formed based on the developed country perspective, which makes it difficult for developing country viewpoints to comprehend social entrepreneurship. Economic, political, and social issues are particularly severe in developing nations. social entrepreneurship. appears to be a successful strategy for finding innovative ways to address these economic and social problems. Thus, we explained several cases from Sri Lanka as a developing country. Thus, the notion of social entrepreneurship is evolving quickly and gaining yet more attention from policymakers and entrepreneurs in developing nations.
Thank you!
Madhuwanthi Premadasa
Reviewer 3 Report
Dear all,
The article utilizes a good strategy and outlines crucial aspects about social entrepreneurship conceptual model which needs to be developed in the developing country context.
The article's organization follows a logical and consistent pattern. The results produced in accordance with the method adopted lead to the conclusions.
The majority of the references are recent publications.
If you consider necessary please include the limitations/ benefits of research when will take in consideration a representative sample and possibility to generalise the results.
Good luck
Author Response
Dear Sir/Madam,
Thank you for your valuable comments and those were supported to enhance the quality of our study. Herewith, we have provided a point-by-point response to the reviewer's comments.
01. The article utilizes a good strategy and outlines crucial aspects about social entrepreneurship conceptual model which needs to be developed in the developing country context.
A good comment
02. The article's organization follows a logical and consistent pattern. The results produced in accordance with the method adopted lead to the conclusions.
A good comment
03. The majority of the references are recent publications.
A good comment
04. If you consider necessary, please include the limitations/ benefits of research when will take in consideration a representative sample and possibility to generalise the results.
As mentioned in the comment, I specified the limitations of the study and future research directions in the later part of the manuscript to enhance the value of the study.
Thank you
Madhuwanthi Premadasa
Reviewer 4 Report
In the introduction, it is necessary to describe the social problems of entrepreneurship in Sri Lanka, so this study is necessary
What qualitative design would phenomenology, grounded theory or case study choose and give reasons
The research method describes the strategy to support the validity and reliability of the research
The proposition in the results and SEO lines 290-325 is a discussion of research results, it should be done after there is a conclusion from the data and the relationship between the themes after line 782.
It is necessary to mention the data from the interview results from which of the 24 participants to make conclusions.
Discuss the results of the research with the previous literature review
Author Response
Dear Sir/Madam,
Thank you for your valuable comments and those were supported to enhance the quality of our study. Herewith, we have provided a point-by-point response to the reviewer's comments.
01. In the introduction, it is necessary to describe the social problems of entrepreneurship in Sri Lanka, so this study is necessary.
As the study is conducted in social entrepreneurship, the authors have added several key social problems faced by Sri Lankan social enterprises in the introduction. Accordingly, it is highlighted the importance of having social entrepreneurship in Sri Lanka as a developing country.
02. What qualitative design would phenomenology, grounded theory or case study choose and give reasons.
Grounded theory can be used to research a certain event or process and come up with novel theories that are supported by data collection and analysis from the real world. Accordingly, in this study, we have used an iterative process for data gathering, data analysis and theory development and propositions for advancing social entrepreneurial orientation in achieving the triple bottom line and model proposed by the study. Thus, a qualitative design used here is grounded theory.
03. The research method describes the strategy to support the validity and reliability of the research.
A good comment
04. The proposition in the results and SEO lines 290-325 is a discussion of research results, it should be done after there is a conclusion from the data and the relationship between the themes after line 782.
Here, after proposing the propositions we have explained the dimensions of social entrepreneurial orientation. Thus, it is difficult to take 290-325 lines after 782. If we change it the flow will be broken. However, we have added a conclusion in the later part of the study.
05. It is necessary to mention the data from the interview results from which of the 24 participants to make conclusions.
Here, we have added the quotes of the 24 respondents to support the discussion and conclusion.
Do we need to submit the transcriptions?
06. Discuss the results of the research with the previous literature review.
I discussed the results with the previous literature. But most of the results are very new and those were supported by the gathered data.
Round 2
Reviewer 1 Report
All comments are duly addressed.
Reviewer 2 Report
The work has improved the aspects discussed in the previous review
Reviewer 4 Report
in general the revisions have improved the quality of the manuscript as a whole
After proposing the theme proposition that emerges from the research results, it is necessary to draw a framework for the proposed model and provide an explanation before the conclusion section